# Recent Progress of Neuromorphic Computing Based on Silicon Photonics: Electronic–Photonic Co-Design, Device, and Architecture

**Bo Xu** [1], **Yuhao Huang** [1], **Yuetong Fang** [1], **Zhongrui Wang** [2], **Shaoliang Yu** [3] and **Renjing Xu** [1,*]

1 Thrust of Microelectronics of Function Hub, The Hong Kong University of Science and Technology (Guangzhou), Nansha, Guangzhou 511400, China
2 Faculty of Engineering, The University of Hong Kong, Hong Kong 999077, China
3 Research Center for Intelligent Optoelectronic Computing, Zhejiang Laboratory, Hangzhou 311121, China
* Correspondence: renjingxu@ust.hk

**Abstract:** The rapid development of neural networks has led to tremendous applications in image segmentation, speech recognition, and medical image diagnosis, etc. Among various hardware implementations of neural networks, silicon photonics is considered one of the most promising approaches due to its CMOS compatibility, accessible integration platforms, mature fabrication techniques, and abundant optical components. In addition, neuromorphic computing based on silicon photonics can provide massively parallel processing and high-speed operations with low power consumption, thus enabling further exploration of neural networks. Here, we focused on the development of neuromorphic computing based on silicon photonics, introducing this field from the perspective of electronic–photonic co-design and presenting the architecture and algorithm theory. Finally, we discussed the prospects and challenges of neuromorphic silicon photonics.

**Keywords:** neuromorphic computing; silicon photonics; optical neural networks; neuromorphic photonics; optoelectronics

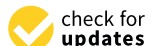



## 1. Introduction

In the past few years, neural networks on conventional computers have been unable to meet the requirements of computational speed and energy consumption due to the limitation of memory walls. In the field of electronic hardware, researchers continue to create deeper and more complex neural network architectures to exploit the potential of electronic hardware platforms [1,2]. As a result, many innovations in processing units can unlock the capabilities of traditional electronic systems. For instance, by aggregating a large number of processing cores, the GPU (graphics processing unit) has exceptionally high parallel computing capability, significantly better than the CPU (central processing unit), which greatly facilitates the development of deep learning [3].

As artificial intelligence evolves, the demand for high performance, energy efficiency, and larger bandwidth in deep learning are endless. As the exponential expansion of electronic transistors, highlighted by Moore's Law, reaches its physical limits, conventional silicon-based electronic components gradually developed a bottleneck of unsustainable performance growth. More underlying electronic components were proposed for the stage of neural networks; memristors [4], phase-change memories (PCMs) [5], ferroelectric random-access memories (FeRAMs) [6], and magnetic random-access memory (MRAMs) [7] are examples of innovative non-volatile memories with high processing speed, huge store capacity, and extended endurance and have been used to demonstrate neuromorphic computing. They can perform efficient neuromorphic computations better than conventional electronic components. Nevertheless, electronic connections fundamentally suffer from harsh trade-offs between bandwidth and interconnectivity, which limits the development of high-speed neuromorphic computing.

In the last few years, photonics has started to gain tremendous attention in academia because of the light-speed data processing and parallel transmission that can be achieved at every level of integrated photonic circuits. Unlike electrons, light has more dimensions, such as wavelength, polarization, and spatial mode, which makes approaches to neuromorphic computing or deep learning more creative and feasible. Furthermore, the mature and advanced technology of silicon photonics provides a perfect platform for large-scale photonic fabrication and integration. Amongst all recent schemes, silicon photonics has been regarded as one of the most promising technologies for neuromorphic computing.

Silicon photonics takes advantage of existing CMOS compatibility; therefore, it can be integrated with available CMOS circuits without the need for additional complex processes. Research on neuromorphic computing based on silicon photonics has made rapid progress [8,9], and various photonic devices have been used for this purpose. Mach-Zehnder Interferometer (MZI), micro-ring resonator (MRR), microcomb, plasma photonic crystal, and phase change materials (PCM) have made optical implementations of optical neural networks or neuromorphic computing based on silicon photonics more feasible and imaginative. In silicon photonics, electronic–photonic co-design is currently one of the most promising schemes for the photonic implementation of neural networks. Moreover, the wide field of emerging applications and devices compatible with silicon photonics, such as MRR, microcomb and PCM, provides the possibility of co-design with more mature electronics. In recent years, numerous architectures and algorithms have been proposed to implement neuromorphic photonic processors, and all these efforts show new possibilities and directions for photonic neural networks.

Here we reviewed the progress made over the past few years in this rapidly evolving field, including the electronic–photonic co-design, devices, architectures, and algorithms for neuromorphic computing based on silicon photonics. First, we investigate the microstructure of neurons from the perspective of electronic–photonic co-design and explore the various trade-offs between all-optical and optoelectronic implementations of neurons. Then, we discussed three types of devices—PCM, soliton microcomb, and metasurface and highlighted their essential roles in this area. Subsequently, we focused on photonic neural networks and corresponding algorithms at the system level. Finally, we provided perspectives for further improvement of neuromorphic computing based on silicon photonics.

## 2. Electronic–Photonic Co-Design

Silicon photonics utilizes basic CMOS fabrication techniques and combines electronic and photonic circuits. Most of the early work on optical neural networks in silicon photonics utilized both optics and electronics. In this section, we reviewed different approaches to implementing micro-architecture functionality with silicon photonic devices and discussed the difference between electronic–photonic co-design and all-optical neuromorphic computing.

### 2.1. Weight

The weighting function is essential to mimic a biological synapse since changing weights is the primary function of learning in a neural network. As learning continues, these parameters are adjusted toward the values that produce the correct output. In silicon photonics, MRRs are a common method for adjusting the weight value and were first employed to implement the weighting function and matrix multiplication [10]. In 2014, a silicon photonic architecture called "broadcast-and-weight" (see Figure 1a) was proposed and demonstrated to realize tunable weighted connections [11].

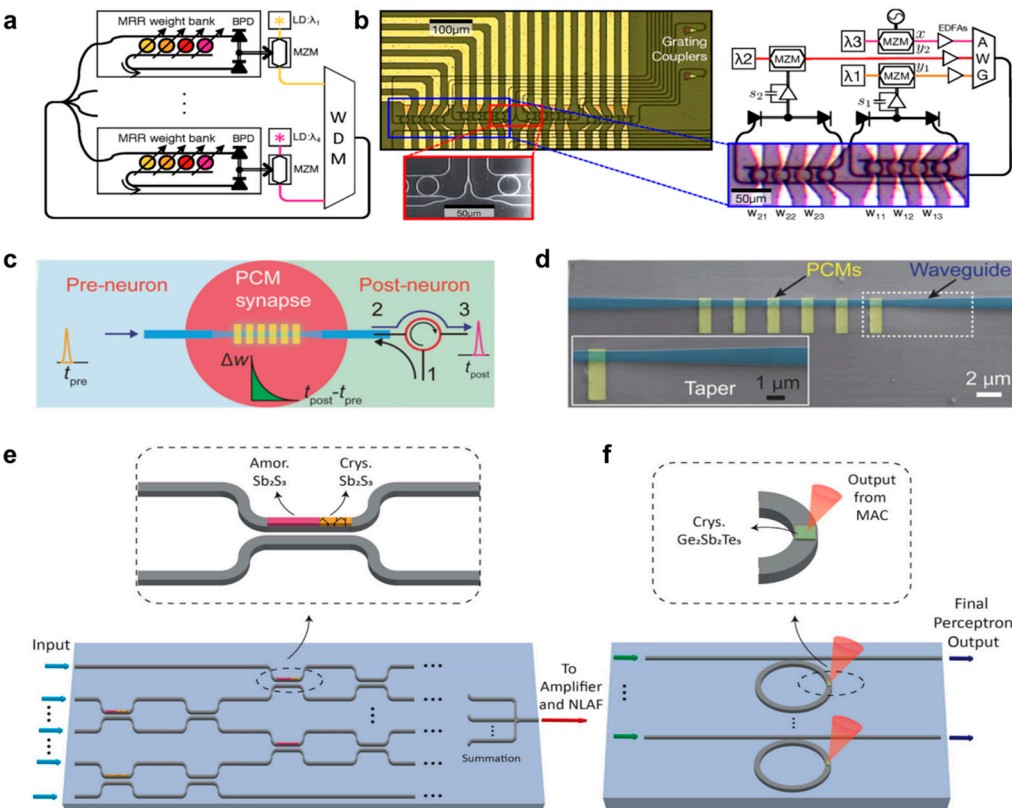

**Figure 1.** Weighting function in ONNs. (**a**) Concept of a broadcast-and-weight network with modulators used as neurons. MRR: microring resonator, BPD: balanced photodiode, LD: laser diode, MZM: Mach–Zehnder modulator, WDM: wavelength-division multiplexer [10]. (**b**) Micrograph of 4-node recurrent broadcast-and-weight network with 16 tunable microring (MRR) weights and fiber-to-chip grating couplers [10]. (**c**) Schematic of the integrated photonic synapse resembling the function of the neural synapse [12]. (**d**) Scanning electron microscope image of the active region of the photonic synapse [12]. (**e**) The weighting and summation mechanisms are based on a cascade of $Sb_2S_3$-SiN hybrid photonic switches that serve the same function as an optical counterpart of an FPGA (Field Programmable Gate Array) [13]. (**f**) The NLAF (Non-linear Activation Function) module consists of a single-mode hybrid silicon waveguide [13]. (**a**,**b**) Adapted with permission from Ref. [10]. CC BY 4.0. (**c**,**d**) Adapted with permission from Ref. [12]. CC BY 4.0. (**e**,**f**) Adapted with permission from Ref. [12] CC BY 4.0.

Devices using phase change materials allow for the storage of up to eight levels of data in a single unit that can be adjusted by light pulses [14]. Chalcogenide $Ge_2Sb_2Te_5$ (GST) is a well-studied phase-change material that has been shown to enable photonic synapses in spiking neurons. Figure 1c shows the scheme for integrating photonic synapses with similar function to neural synapses, which contains multiple PCM islands for precise control of synaptic weights [12]. This study also demonstrates that employing PCM islands in combination with a tapered waveguide structure is more effective than using a regular non-tapered waveguide. However, the application of GST is limited by its high optical absorption and phase transition time. In order to overcome the high optical absorption of GST, Ting Yu et al. adopted $Sb_2S_3$ to set the synaptic weights [15]. After the linear weighted addition, the optical signals interact with the GST layer nonlinearly, as shown in Figure 1e,f. Their results show the advantage of noise robustness and potential sub-picosecond delay of embedding this material on a silicon photonics neural network platform.

In summary, the implementation of the weighting function in silicon photonics without electro-optical conversion relies on non-volatile materials. The advantages of using all-optical weighting functions without electrical–optical conversions are evident in many

ways, such as high bandwidth and high speed. Nevertheless, optical weight in the photonic synapses suffers from the cost of a large area, which is one of the key indicators for evaluating the practicality of optical schemes. For instance, a conventional memristor-based synapse can be as tiny as 10 nm when implemented electrically [16], whereas the suggested synapse in reference [12] is as huge as 6 μm by 1 μm. Compared to all-optical methods, neurons with the same functionality and precision can differ in the area by three or four orders of magnitude using the electrical method.

### 2.2. Summation

In addition to the straightforward sum of input and weights, the summation function can be more complex; for example, it can also be a minimum, maximum, majority, product or several normalization procedures. The selected network design and paradigm determine the exact algorithm for merging neural inputs, and it is easy to implement the summation function in a software method. However, the summation function is significant in an optical artificial neural network since it always affects the scalability. In most previous work, summation requires signal conversion because photodetectors are essential for combining inputs in the function [10]; therefore, the capability of micropillar semiconductor lasers, DFB lasers, and VCSELs was investigated in order to implement the summation function optically [17].

PCM-based implementation can support noncoherent neuron summation functions in the all-optical domain [18]; however, the scalability is limited by current all-optical methods since the optical summation functions are not compatible with WDM. Further, all-optical implementation is required to support a large number of optical inputs at the same time, where photodetectors, as the mainstream method, can provide a simple implementation but still suffer from low reliability, high noise sensitivity, and low integration.

### 2.3. Activation

The activation function can be materialized by using electronic or photonic methods, while the effective expressiveness of the artificial neural network depends on the nonlinearity of the activation function.

Currently, the photonic activation function is still in its preliminary stage. For instance, some researchers are still realizing it through digital computers and feeding the modulated optical signals to the next layer. In such OEO (optical–electrical–optical) activation function, the frequent electro–optical conversion processes lead to a large number of delays and power consumption, which limits the speed of neural networks. Therefore, Ian A. D. Williamson et al. proposed hardware nonlinear optical activation, which does not require repeated bidirectional optoelectronic signal conversions, as shown in Figure 2a. In this activation function, most of the signal power remains in the optical domain because it converts only a small portion of the input optical signal into the analog electric signal to intensity-modulate the original optical signal [19]. In addition to this, optical modulators by using a combined structure of MRR [20] and SOA [21] are also suitable for realizing nonlinear activation; for example, as shown in Figure 2b, the proposed layouts exploit a Semiconductor Optical-Amplifier (SOA)-based sigmoid activation within a fiber loop. Vertical-cavity surface-emitting lasers (VCSELs) are also great candidates for activation functions, and they exhibit many profound advantages [22], such as being easy to integrate into 2D or 3D arrays, low power consumption, and coupling efficiency to optical fibers. In reference [23], J. Robertson et al. discussed experimentally and theoretically the dynamics of spikes in VCSELs with respect to controlled suppression. Furthermore, according to reference [24], excitation and suppression can be achieved in VCSELs using dual-polarization injection.

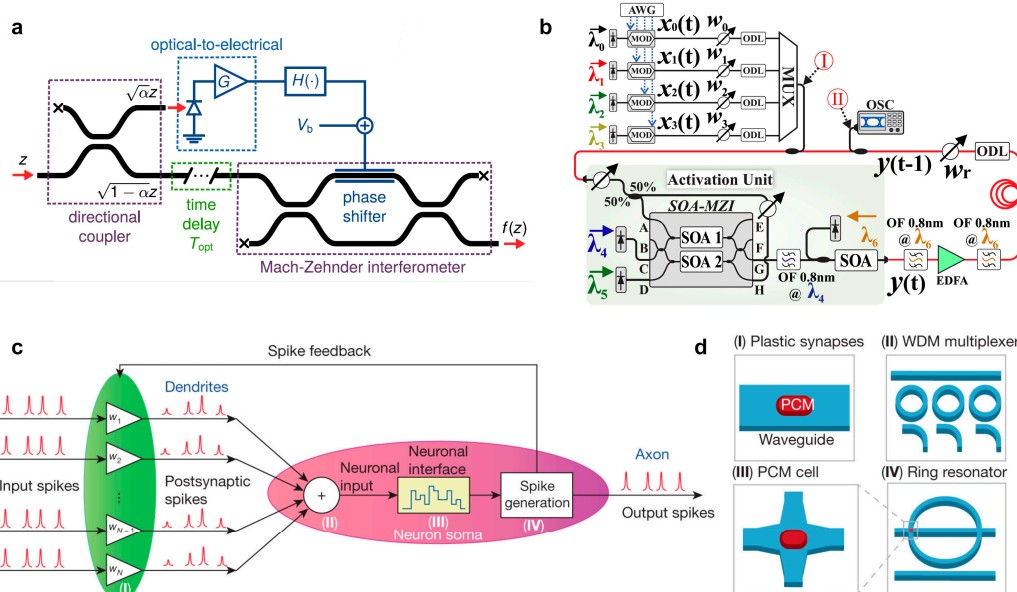

**Figure 2.** Activation function in ONNs. (**a**) Schematic of the proposed optical-to-optical activation function [19]. (**b**) Experimental setup used for the evaluation of the 4-input WDM RNN (Recurrent neural network) [21]. (**c**,**d**) Schematic of the network realized in this study, consisting of several pre-synaptic input neurons and one post-synaptic output neuron connected via PCM synapses [18]. (**a**) Reprinted with permission from Ref. [19]. Copyright © 2020, IEEE. (**b**) Reprinted with permission from Ref. [21]. Copyright © 2020, IEEE. (**c**,**d**) Reprinted with permission from Ref. [18], Springer Nature Limited.

Figure 2c,d shows a PCM-based all-optical neuron [18], where PCM is placed on the MR, resembling the activation unit. Due to changes introduced in the structure of the PCM on the waveguide crossing, the transmission response undergoes a considerable change to obtain the ReLU function.

### 2.4. STDP

The learning capability is required to build neural networks in both spiking and traditional neurons. In spiking neurons, Spiking Time Dependent Plasticity (STDP) learning is inherently asynchronous and online, which contrasts with the more conventional learning functions; STDP changes the weight of the synapse according to the precise timing of individual pre-synaptic spikes and post-synaptic spikes [25]. Functionally, STDP serves as a mechanism that performs the Hebbian learning rule that the strength of neural connections is determined by the correlations between pre- and post-synaptic activity. SOAs and MRRs are both promising candidates for implementing STDP learning. A photonics approach to implement STDP using SOA and Electro-Absorption Modulator (EAM) was introduced experimentally in reference [26]. STDP-based unsupervised learning is also theoretically explored in principal component analysis (PCA). Moreover, high-order passive MRRs were demonstrated to be an alternative photonic STDP approach [27], and since the proposed scheme is passive, it has a lower power consumption compared to schemes using SOA, where the intracavity effect induces a power difference at the output of the MRR from which the inter-spike is calculated.

## 2.5. All-Optical versus Optoelectronic Neurons Implementation

As discussed previously, most previous work on ONNs was implemented based on optoelectronic hardware [28]. However, in optoelectronic hardware, massive power is consumed in the electrical-to-optical and its inverse conversion because the device works more in the electrical domain in the summation and activation functions. For instance, photodetectors are frequently used to convert optical signals to electrical outputs, which imposes restrictions on the speed and power efficiency of ONNs. Moreover, O/E/O neurons rely on modulators that utilize the nonlinearity of the electro-optic transfer function. Nevertheless, modulators and photodetectors are susceptible to noise, and their noise accumulation can seriously affect the accuracy and energy consumption of ONNs based on optoelectronic hybrid hardware.

All-optical implementation seems to be a promising approach to addressing the problems of optoelectrical hybrid hardware. Compared to the electronic implementation, all-optical neurons usually rely on the semiconductor carriers or photosensitivity that occur in many materials. The most obvious advantage is that the optical signal flow in all-optical neurons does not require any conversion; therefore, they are inherently faster than O/E/O schemes. Meanwhile, all-optical schemes using passive optical components can be easily integrated with CMOS technology. The photonic implementation also provides the advantages of the high bandwidth in photonic communication and the low complexity in nonlinearity implementation. However, many challenges remain with all-optical neurons, and cascadability is a key challenge for photonic implementations. Designing all-optical neurons requires more efficient optical devices due to their insertion loss; nevertheless, cascadability must compensate for the power consumption at the level of the system in some all-optical implementations [29].

From a more practical perspective, electronic and photonic co-design hardware enables neuromorphic computing, which is one of the core research routes until the required optical devices leap over the current challenges. Nevertheless, the gap between electronics and photonics neural networks has always existed because most architectures are designed for a specific platform rather than for optical hardware. Moreover, photonics endows the traditional neural networks in the electrical domain with many more unique benefits that are difficult or even unable to implement by using electrical devices, even though we acknowledge that the capabilities of ONNs lag far behind the electrical neural networks. That is one of the most core reasons for us to conduct massive research on ONNs and those optical components.

Here, we propose a possible framework for the optoelectronic-hybrid AI computing chip that can be accessible in a reachable position nowadays. As shown in Figure 3, the framework consists of three parts: an optical engine, an analog electronic part, and a digital ASIC or DSP. The optical engine and DSP require some analog electronic components in order to interact with one another because they function with signals of different strengths. The DSP sends signals to the electrical driver block, which amplifies them and uses them to power the optical engine. The DSP or ASIC chip has a series of processing units that recover, decode, and error-correct the data streams after compensating for various transmission problems. Various applications might need slightly different DSP layouts or may not need all of the processing modules. A far better fit between these parts can be achieved by co-designing the photonic integrated circuit (PIC) and the DSP chip. The trade-offs between different DSP and PIC characteristics can be more precisely identified with the use of a co-design technique, which also enhances system-level performance optimization.

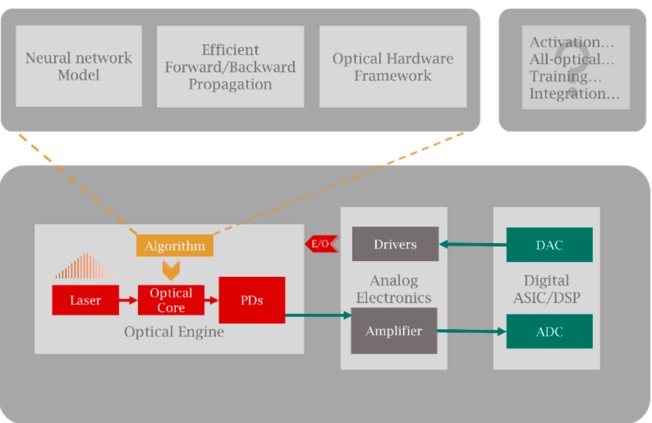

**Figure 3.** A possible framework for optoelectronic-hybrid AI computing.

## 3. Devices

In this section, we reviewed some important devices based on solitons, PCM, and metasurfaces that can be employed to implement ONNs. As we showed before, MRRs and photodetectors are used to achieve electrical to optical conversion and optical to electrical conversion, respectively. In addition to these three kinds of devices, other components based on silicon photonics, such as lasers, couplers, and modulators, are also critical parts of optical circuits and neural networks. As for on-chip lasers, both vertical Cavity Surface Emitting Lasers (VCSEL) and microdisk lasers support the design of scalable neural networks, although many researchers still employ off-chip lasers to constitute neural networks. Moreover, waveguides are of great importance in silicon photonics because they are equivalent to metal wires in the electrical domain. How to minimize optical loss, including propagation loss and bending loss, is under massive research. In addition, MRRs, microdisks, and MZIs are widely employed to design modulators, switches, and filters [9]. Although silicon photonics can now be considered a mature technology platform compared to optical neural networks, the issue of connecting light to and from silicon photonic components with high efficiency remains a challenge.

Recently, devices that utilize PCM, soliton microcombs, and metasurfaces have been of great interest in photonic neuromorphic computing. In reference [30], the concept of time–wavelength multiplexing for ONNs was introduced, and the Kerr microcomb was applied to implement a photonic perceptron. In 2021, Xu et al. demonstrated a universal optical vector convolutional accelerator based on simultaneously interleaving temporal, wavelength, and spatial dimensions enabled by an integrated microcomb source [8]. Meanwhile, Feldmann et al. proposed a scheme for an integrated photonic tensor core using phase-change-material memory arrays and soliton microcombs [31]. Moreover, metasurfaces demonstrate a different dimension for on-chip optical neural networks [32]. Metasurfaces with subwavelength resonators are used to manipulate the wavefront of light, allowing the miniaturization of free-space and bulky systems for diffractive neural networks (DNN). By using soliton microcombs, PCMs, and metasurfaces, these novel methods based on silicon photonics enable optical neuromorphic computing, providing an effective way to break through previous bottlenecks in machine learning in electronics.

### 3.1. Soliton Microcombs

#### 3.1.1. Basic Science

Optical Frequency Combs (OFCs) have traditionally been built on the mode-locked mechanism or fiber lasers; however, recent advances have demonstrated an OFC generated in a Kerr-nonlinear optical microresonator, commonly referred to as microcombs. A partial history of the development of optical of the optical frequency comb can be obtained from Figure 4. A microcomb depends mainly on dissipative Kerr solitons (DKS), and solitons are stable waveforms that can retain their shape when propagating within a dispersive

medium [33]. Optical solitons have particle properties, and multiple solitons can form a variety of bound states through interactions. On a fundamental level, solitons can be found in a plethora of nonlinear systems. In terms of nonlinear dynamics and phenomena, DKSs as a type of solitons have attracted significant attention and demonstrated a wide range of applications such as RF photonics, parallel coherent LiDAR, optical frequency synthesizer, and photonic neuromorphic computing [34–36]. As shown in Figure 5a, a microresonator driven by the CW pump is widely used to build a Kerr soliton microcomb. By balancing the loss and gain in the active medium while balancing the nonlinearity and dispersion and remaining in the anomalous group velocity dispersion (GVD) regime, the high-Q microresonators can support fully mode-locked comb states called DKS [37]. Figure 5b shows the interactions of Kerr combs with other coexistent nonlinear effects, such as nonlinear photon scattering and second-order nonlinear processes, which are associated with the light-matter interaction in high-Q microcavity platforms [38].

Microcombs are still poorly studied for photonic computing; however, recent advances in microcombs show they could be a promising candidate as a chip-scale light source. Kerr-nonlinearity induced by optical parametric oscillation in a microcavity was first reported in 2004 [39]. Figure 4 illustrates a significant development in soliton microcombs. In 2003, the first silicon laser using a silicon waveguide as a gain medium was demonstrated [40]. Then, in 2005, Alexander W. Fang et al. reported the first electrically pumped hybrid silicon laser [41]. As an indispensable component of a fully integrated silicon photonic circuit, recent research on on-chip silicon lasers has also paved the way for future integrated neuromorphic computing circuits [42]. In parallel, in 2007, an optical frequency comb using an approach different from the comb-like mode structure of mode-locked lasers was first experimentally generated by the interaction between a continuous-wave pump laser of a known frequency with the modes of a monolithic ultra-high-Q microresonator via the Kerr nonlinearity [43]. In reference [44], a method for the generation of soliton microcomb was first proposed and is used by academics to date. By artificially scanning the pump frequency or the cavity resonance from blue to red (or from red to blue using a heater for resonance), the pump laser can reach the DKSs region and thus generate the soliton microcombs. Recently, the integrated turnkey soliton microcomb was demonstrated and theoretically explained [45]. The microcomb co-integrated with a pump laser operates at CMOS frequencies as low as 15 GHz with optical isolation removed, providing significant advantages for high volume production, and this integration can also be exploited by integrated neuromorphic computing circuits, as we discussed previously [45].

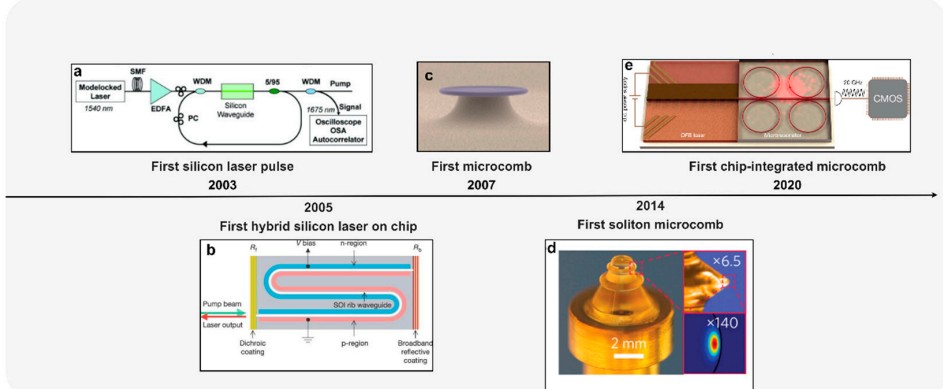

**Figure 4.** The development history of microcombs and some significant events. (**a**) Reprinted with permission from Ref. [40], ©2004 Optica Publishing Group. (**b**) Reprinted with permission from Ref. [42], Copyright © 2005, Springer Nature Limited. (**c**) Reprinted with permission from Ref. [43]. Copyright © 2007, Nature Publishing Group. (**d**) Reprinted with permission from [44]. Copyright © 2019, Springer Nature Limited. (**e**) Reprinted with permission from [45]. Copyright © 2020, Springer Nature Limited.

### 3.1.2. Computing Based on Soliton Microcomb

Recent research has shown the potential to leverage soliton microcombs while implementing the neural network [8,31]. Prior to the use of soliton microcombs, different multiplexing methods to realize parallel synapses successfully demonstrated their positive application in ONNs. For example, photonic reservoir computing employs time-domain multiplexing to build large-scale input layers with many nodes [46]. However, the training and scalability of time-division multiplexed networks are limited in the current scheme, whereas the introduction of microcomb sources into ONNs can combine the use of wavelength, time and spatial multiplexing simultaneously, offering benefits to the matters discussed before.

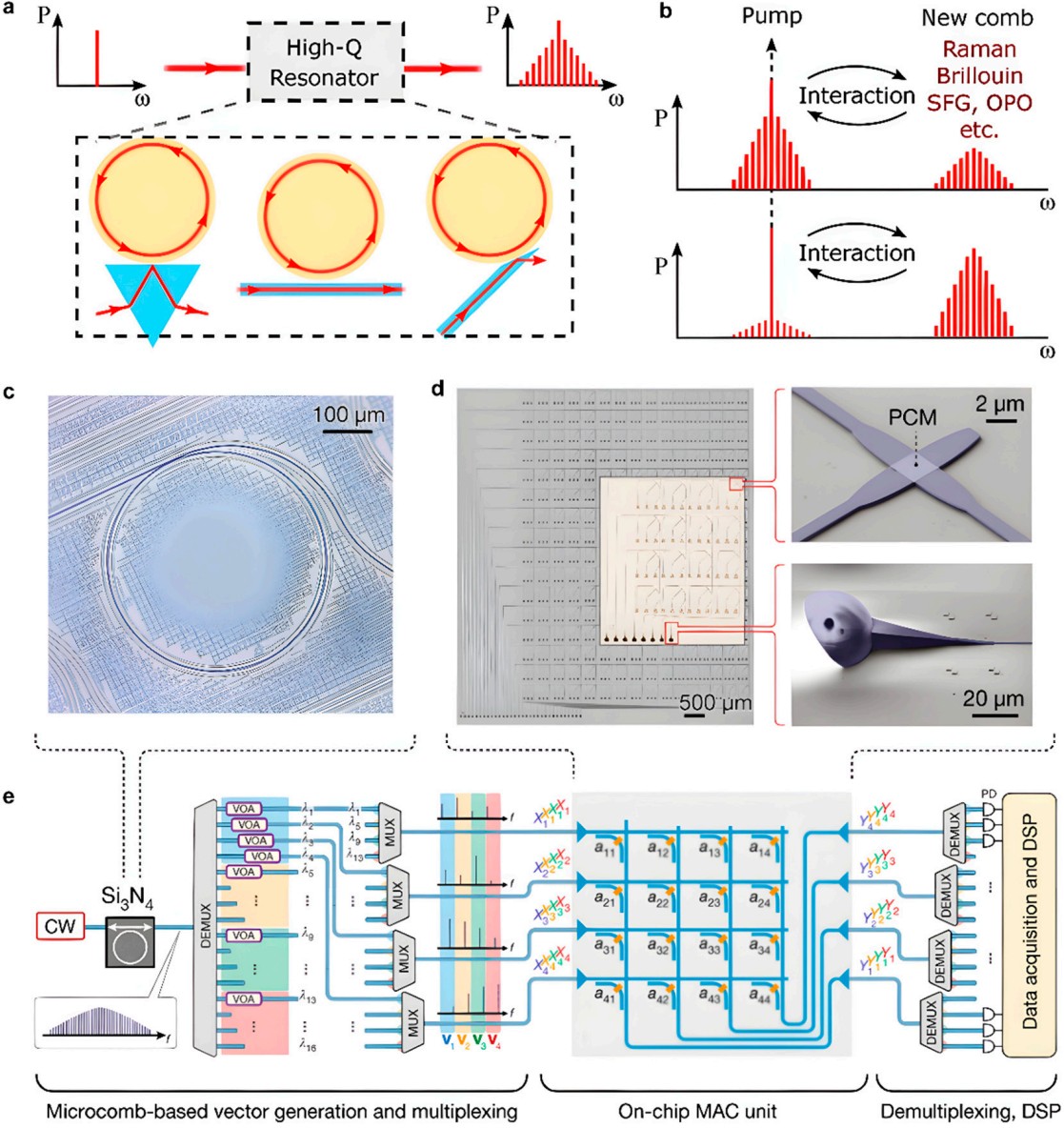

**Figure 5.** Computing Based on Soliton Microcomb. (**a**) Overview of high-Q resonator platforms for the generation of Kerr frequency combs [38]. (**b**) Illustration of Kerr comb interaction with other cubic and quadratic nonlinear effects [38]. (**c**) Sketch of the multiplexed all-optical MVM [31]. (**d**) Optical micrograph of a high-Q $Si_3N_4$ photonic-chip-based microresonator used for frequency comb generation [31]. (**e**) Optical micrograph of a fabricated 16 × 16 [31]. (**a,b**) Reprinted with permission from Ref. [38]. © 2021 Wiley-VCH GmbH. (**c–e**) Reprinted with permission from Ref. [31]. Copyright 2021, The author(s), under exclusive license to Springer Nature Limited.

In reference [30], a single perceptron based on soliton microcombs was reported operating at 11 billion ($10^9$) operations per second, in which the perceptron has 49 synapses. Based on wavelength multiplexing with 49 microcomb wavelengths, simultaneously with temporal multiplexing, the input nodes of the perceptron differ from conventional ONNs because they are temporally defined by multiplexing the symbols that are then routed, determined by their location in time. By following this work, Xu et al. demonstrated a photonic convolutional accelerator (CA) with the concept of time–wavelength multiplexing [8], where they combined the CA front end and a fully connected neuron layer to form an optical CNN. The chromatic dispersion is used to employ time delay to the wavelength-multiplexed optical signals, which are then combined along the dimension. By using the microcomb source, the CA is capable of speeds up to 11.3222 TOPS, and it can process 250,000-pixel images using a single processing core.

Feldmann et al. made an integrated photonic hardware accelerator using the phase-change material memory and soliton microcombs [31], as shown in Figure 5e, where the processor encodes data onto the on-chip microcomb teeth and performs matrix–vector multiplication using the non-volatile configuration of the PCM's array of integration units. In this approach, the hybrid integrated microcomb and PCMs integrated onto waveguides enable in-memory photonic computing using WDM capability; moreover, the parallelized implementation is CMOS-compatible and promises higher bandwidth, lower power, and real speed of light since the convolutional operation is a passive transmission measurement, thus making it possible to process the entire image in a single step.

Combined with recent advances in soliton microcomb, ONNs fully integrated on a chip is potentially viable. However, in addition to soliton microcombs, solitons can be exploited to implement more computing systems. For example, a variety of logic devices such as half-subtractor, comparator, and logic AND, OR, XOR, and NOT gates can be implemented by solitons due to their elastic interaction and capability of controlling [47,48]. Nevertheless, these implementations are inherently digital computing; instead, the approaches would throw away the advantages of photonics in analog computing. Moreover, reference [49] theoretically proposed a soliton-based reservoir computing scheme, where they explored the possibility of using interacting soliton chains to work as a reservoir.

### 3.2. Devices Based on PCM

Devices combined with PCMs are of great interest in building advanced neural networks based on silicon photonics. As shown in Figure 6, the past decades have witnessed the development of PCMs. At present, PCMs are one of the most mature and widely investigated materials. Among emerging non-volatile memory devices, PCM-based optical neural networks are promising to address the limitations of electrical neuromorphic computing.

### 3.2.1. Basic Science

The basic principle of PCM devices is to induce phase change between the amorphous and crystalline states of PCM by employing thermally/optically/electrically pulses, during which the conductivity of PCM also switches between low electrical ones (amorphous states) and high electrical ones (crystalline states), corresponding to the low reflectivity state (HRS) and high reflectivity state (LRS), respectively. The crystallization process is generally referred to as SET, which requires PCM to be exposed to a long (hundreds of nanoseconds) voltage or laser pulse. In order to reset the PCM or achieve the amorphization process, short high lasers or electrical pulses sufficient to melt the phase-change layer can be used. Due to these properties above, PCMs were first used as storage mediums for rewritable compact disks such as high-density digital versatile disks (HD-DVD) or blue-ray disks and RAM, as shown in Figure 6.

However, the introduction of PCM into photonic neuromorphic computing poses new challenges to the majority of PCMs, namely Ge-Te-Sb (GST), which are selected according to the criteria of low electrical resistivity and high optical absorption because of low optical loss and high phase change speed obtain more critical in nanophotonic waveguides due to

the proximity of the optical mode to the PCMs. Recent proposed PCMs such as GeTe [50], GSST [51], $Sb_2Se_3$, and $Sb_2S_3$ [52,53] can address the emerging problems in the application of photonic PCM; in particular, $Sb_2Se_3$ and $Sb_2S_3$ can offer near zero loss in both amorphous and crystalline states [54].

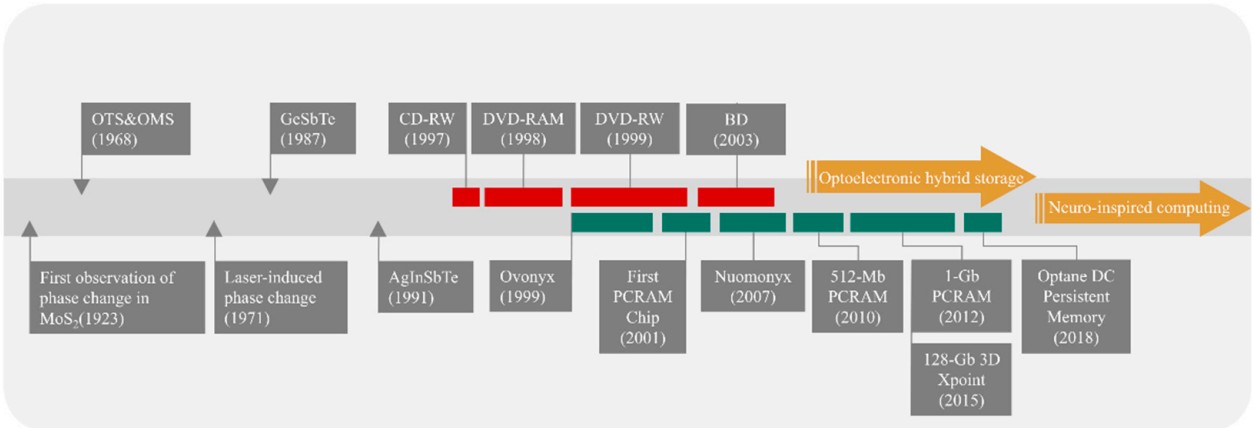

**Figure 6.** The development history of phase change materials.

3.2.2. Phase Change Materials for Integrated Photonics Computing

As we discussed previously, inducing the high refractive index changes to achieve phase tuning of PCMs is the main criterion for employing PCM in photonic devices, while non-volatile tuning allows for low energy overhead in silicon photonic devices.

Phase change memory as the forerunner of the application of phase change materials has a long history since it was first reported by Yamada and his colleagues [55]. In 2012, Penrice et al. demonstrated an all-optical phase change memory by depositing a GST film on a $Si_3N_4$ ring resonator structure on silicon, as shown in Figure 7a. Later, Rios et al. performed a complete analysis of GST memory devices [56] and showed that the resonance wavelength, the Q-factor, and the extinction ratio could be used to retrieve the state of GST, as shown in Figure 7d,e. These two works paved the way for future research on phase change memory. In recent years, $Si_3N_4$ multiple ring resonators with GST patches coupled to a single waveguide, the usage of PWM for switching of PCM, and numerous novel methods for implementing phase change memory have been presented one after another.

After the first report of the phase change memory, the possibility of applying PCM to neuromorphic computation was investigated. Subsequently, PCM has been continuously explored and utilized to implement neuromorphic computing [56–58]. A method for creating scalable PCM-based optical synaptic networks using silicon-based ring resonators with GST patches was proposed [59], which differs from the scheme of creating PCM synapses on a single waveguide [12] (as shown in Figure 1c). By incorporating wavelength division multiplexing better, Feldmann designed the spiking synaptic network to achieve both supervised and unsupervised self-learning, as described above, also using a waveguide implementation with a PCM integrated on top [18].

In order to reconfigure phase-change photonic devices, it is frequently necessary to adjust the intensity [14] and pulse shape [60] of an incident light wave. Resonant structures [14] are frequently utilized to enable the wavelength-selective operation and improved modulation depths. However, the comparable feature is not present in polarization space, which limits the addressability of distinct pieces in cascaded systems. Hence, June et al. proposed a hybridized-active-dielectric structure in a nanowire configuration, in which the GST as the active material undergoes a phase change by sending power- and polarization-modulated laser pulses and silicon acts as dielectric [61]. Polarization-selective dielectric resonances of the Si cavity modify the total absorption in the GST layer. With up to five independent levels of reconfigurable, non-volatile polarization-division demultiplexing of electrical conductivity, they demonstrate matrix–vector multiplication based on the

hybridized-active-dielectric structure (MAC-type operations) with input polarization as the tunable vector element, which unlocks an additional route in phase-change photonics.

Given the negative properties of GST in photonic PCM networks, GSST, $Sb_2S_3$, and other alternative materials were produced to address the problems. In 2021, Yu and his colleagues used $Sb_2S_3$ instead of GST in some crucial parts of networks, such as the weighting function [15]. They used two different transitions within chalcogenide in optical synapse weighting and all-optical nonlinear thresholding to enable a low loss and high-speed all-optical deep neural network. GSST was also demonstrated as a promising candidate. In reference [62], Volker utilized MZIs with GSST on both arms, using the electrical switching method instead of the optical approach. Recently, the integrated photonic tensor core for parallel convolutional processing based on GST demonstrated a scalable, high-speed ($10^{12}$ MAC per second) and low-power (17 fJ/MAC) scheme by using the integrated microcomb as we discussed previously [31].

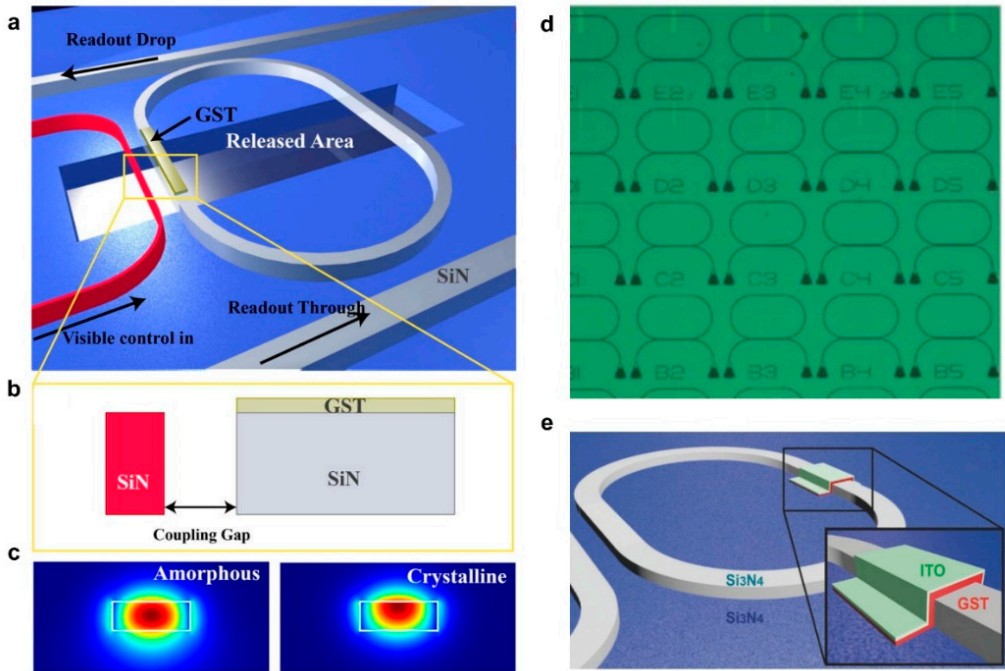

**Figure 7.** Phase change materials for integrated photonics computing. (**a–c**) Schematic overview of the proposed memory element [63]. (**d**) Optical microscopy image of an array with 25 different memory elements [64]. (**e**) Three-dimensional scheme of the platform using partially etched ridge waveguides in silicon [64]. (**a–c**) Reprinted with permission from Ref. [63], AIP Publishing. (**d,e**) Reprinted with permission from Ref. [64]. © 2013 WILEY-VCH Verlag GmbH & Co. KGaA, Weinheim.

### 3.3. Metasurfaces

Metasurfaces consisting of subwavelength-spaced building blocks have inhomogeneously distributed optical resonators, and optical wavefront can be reshaped and redirected by the abrupt phase shift caused by resonators to generate phase discontinuities when light passes through them. Optical computing benefits from metasurfaces since they provide multiparametric optical modulation in a single element, which is competitive and promising to miniaturize and integrate bulky free-space optical systems, such as optical integrators [65], differentiators [66], and diffractive neural networks [67].

#### 3.3.1. Basic Science

Metasurfaces are two-dimensional metamaterials, which were earlier studied by designing the meta-atoms and their different spatial arrangement with the aim of generating abnormal effective medium coefficients and investigating the accompanying aberrant

physical phenomena [68]. However, metasurfaces are gradually substituting metamaterials as a research hotspot because they are compatible with standard lithography and nanoimprinting techniques and can be easily tuned dynamically.

As we discussed above, metasurfaces consist of hundreds of subwavelength structure units, each of which generates a spherical wave packet that forms a new electromagnetic wavefront at the metasurface–air interface [69]. Hence, by adjusting the shape, size, and direction of the metasurface unit, metasurfaces can allow ultimate control over beam propagation, divergence, and information encoding [70]. The limitation also lies in the wave-manipulation functionalities since metasurfaces are fixed after design and fabrication, and in this context, progressive research has been devoted to addressing this issue in order to implement reconfigurable metasurfaces upon external tuning [71,72]. As shown in Figure 8, electronic control [73], light control [74], temperature control [75], mechanical control, and power control are the common methods employed by tunable metasurface structures [71]. Moreover, multifunctionality is also a challenge to metasurfaces, which can be addressed by segmented or interleaved metasurfaces [76,77] and by adding complex transmission profiles using the linear property of the Fourier transform [78].

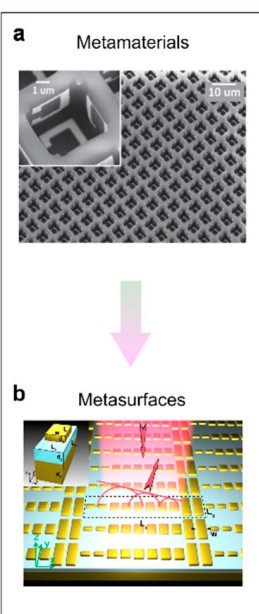
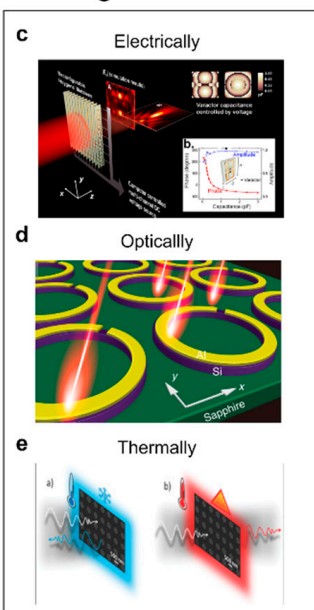
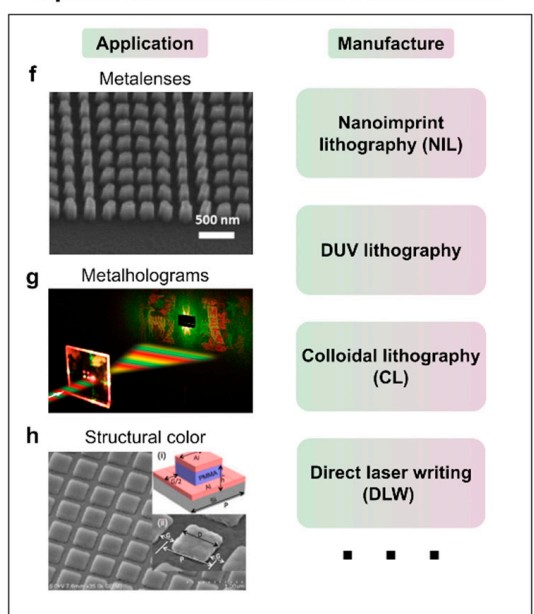

**Figure 8.** Basic science of metasurfaces. (**a**) Reprinted with permission from Ref. [69]. Copyright © 2010 WILEY-VCH Verlag GmbH Co. KGaA, Weinheim. (**b**) Reprinted with permission from Ref. [70]. Copyright © 2012, American Chemical Society. (**c**) Reprinted with permission from Ref. [73]. © 2017 WILEY-VCH Verlag GmbH Co. KGaA, Weinheim. (**d**) Adapted from Ref. [74]. CC BY 4.0. (**e**) Reprinted with permission from Ref. [75]. © 2017 WILEY-VCH Verlag GmbH Co. KGaA, Weinheim. (**f**) Reprinted with permission from Ref. [79], JoVE. (**g**) Reprinted with permission from Ref. [80]. CC BY 4.0. (**h**) Reprinted with permission from Ref. [81]. CC BY 4.0.

Optical metasurfaces can operate in visible and near-infrared spectrums. Because electromagnetic waves in this wavelength range are commonly used in imaging applications, optical metasurfaces in this band are of greater interest than those operating in the mid- and far-infrared wavelength ranges [82]. Moreover, the feasibility of low-cost scalable manufacturing of metasurfaces, depending on their specific design, such as NIL, DUV, and CL, provides a promising prospect for the application of metasurfaces [79–81].

### 3.3.2. Computing Based on Metasurfaces

Metasurfaces can be used to build a meta-system that implements more complex applications, such as optical analog computing and diffractive deep neural network (D$^2$NN),

in both free-space and integrated on-chip ways. Here, we mainly talked about integrated metasurface systems because the metasurface design for integrated photonics platforms follows the same guidelines as free-space metasurface design.

Integrated metasurfaces enable optical analog computing by exploiting the Fourier transform properties. For instance, the light propagation in planar waveguides can be controlled by in-plane one-dimensional metasurface or, in other words, metaline, and be leveraged to implement mathematical operations. Figure 9a shows a three-layer meta-system that can perform the spatial differentiation of the input signal, and Figure 9b demonstrates the on-chip metalens structure within the system [83]. Moreover, the 1D metalens has a numerical aperture of up to 2.14, allowing light to be focused to within 10 m with less than 1 dB loss. In reference [84], on-chip convolution operation was demonstrated based on silicon metasurface with the help of inverse design optimization, for which the proposed convolver performed spatial convolution on the provided function at two wavelengths of 1000 and 1550 nm.

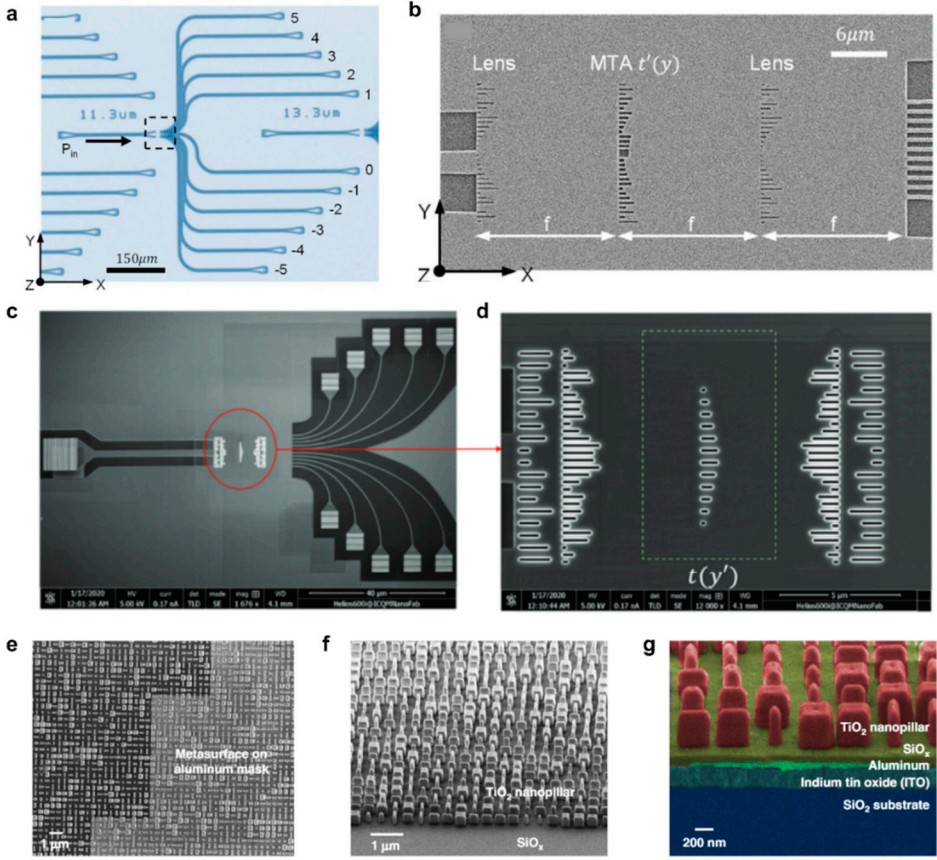

**Figure 9.** Computing Based on Metasurfaces. (**a**) Top view of an on-chip metalens structure. The input light is fed through the waveguide on the left. As the light passes through the microlens, the output is collected through single-mode waveguides at different spatial locations [83]. (**b**) The SEM image of the three-layer meta-system can perform the spatial differentiation of the input signal [83]. (**c**,**d**) Overall view of the SEM image of the on-chip nanophotonic convolver and characteristic structure of this device [84]. (**e**–**g**) Top-view, oblique-view, and false-color cross-sectional view of the scanning electron microscope (SEM) images, respectively, of the MDNN [32]. (**a**,**b**) Adapted from Ref. [83]. CC BY 4.0. (**c**,**d**) Adapted from Ref. [84]**.** CC BY 4.0. (**e**–**g**) Adapted from Ref. [32]. CC BY 4.0.

Metasurfaces also provide an exceptional solution to the issue of miniature integration of ONNs. Early studies have shown several methods to implement diffractive deep neural networks (D$^2$NN) based on free-space optics [85]. The integrated metasystems require less power consumption and footprint and also allow for more information capacity due to the

multimode nature of the diffraction. Zarei et al. demonstrated an integrated photonic neural network based on on-chip cascaded one-dimensional (1D) metasurfaces at a wavelength of 1.55 μm [86]. Problems exist in many ways, such as the diverse, effective refractive index of the same slot at different positions for different angles of light input and mutual interference between adjacent slots of different lengths; thus, to address these issues, an optical deep learning framework was proposed [87]. In this framework, the pre-trained neuron values are physically translated into the various phase delays, and the relevant phase delays are created by altering the size of the silicon slots. In combination with phase-change materials, metasurfaces can enable a prototypical optical convolutional neural network that utilizes the phase transition of the GST to control the waveguide spatial modes with a very high precision of up to 64 levels in modal contrast [88].

Two-dimensional metasurfaces were also used to build on-chip integrated neuromorphic systems [32], where the metasurface devices use a polarization multiplexing scheme and can perform on-chip multi-channel sensing and multitasking in the visible. This proposed method provides a promising avenue for expanding the field of optical neural networks because many multiplexing schemes of the metasurface, such as polarization, wavelength, vortex, etc., can be endowed to all-optical neural networks.

## 4. Architecture and Algorithm

### 4.1. Implementation by Interference of Light

Mach–Zehnder interferometer (MZIs), which are extensively utilized in optical modulators [89], optical communication [90], and photonic computing [91], are crucial components of the optical interference-based network. By using attenuators and phase shifters on the MZI arms to control the weights to change the phase and amplitude of the optical signal, the MZI can act as a natural matrix multiplication unit without fundamental losses.

In 2017, Shen et al. utilized Singular Value Decomposition (SVD)-based methodology to implement the coherent optical computing architecture [28], as shown in Figure 10a,b. Although SVD-based matrix multiplication was demonstrated by reference [92], this was the first practical and advanced demonstration using a Photonic Integrated Circuit (PIC), where the layers of this architecture consist of an Optical Interference Unit (a cascaded MZI mesh using SVD) and an Optical Nonlinearity Unit (implemented with digital electronics).

Although MZIs have shown promising potential in overcoming the bottlenecks of state-of-the-art electronics, it still suffers from a larger footprint than their counterparts and the accumulation of phase errors; hence, different approaches were recently proposed to reduce the area cost of MZI mesh. The author of reference [93] designed a slimmed architecture using a sparse tree network block, a single unitary block, and a diagonal block for each neural network layer, as shown in Figure 10c. They co-designed the optical hardware and software training implementation and achieved area savings of 15–38% for MZIs-based ONNs of various sizes. Another work demonstrated a similar idea that uses the fast Fourier transform (FFT) to reduce area cost [94], see Figure 10d, where the structured neural network used is friendly to hardware implementation while greatly reducing the complexity of the computation, offering motivation to prune SVD-based ONNs. In this work, they used OFFT and its inverse (OIFFT) to implement a structured neural network with circulant matrix representation and pruned the weight matrices by Group Lasso Regularization.

In silicon photonics, it is less common to implement noncoherent architectures by MZIs, while in optical fiber systems, MZIs always work as intensity modulators [95], thus allowing the construction of a recurrent neural network working in combination with an arbitrary waveform generator. Back to the silicon photonics, RNNs can likewise be built using the SOA-MZI units, as shown in Figure 2b, in which the SOA is placed on the arms of MZIs, thus performing as a wavelength converter for cross-gain modulation [21] and enabling an all-optical WDM functionality.

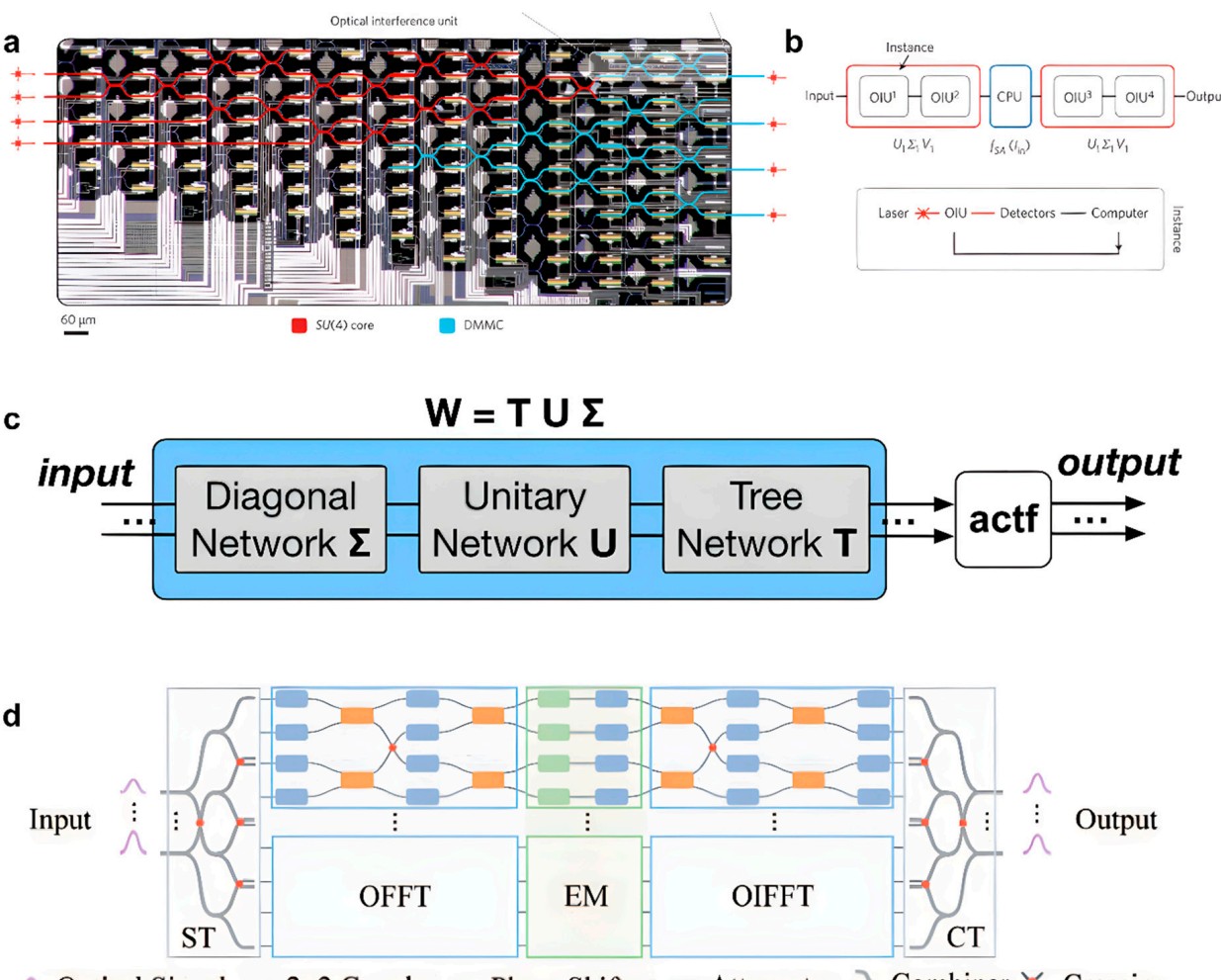

**Figure 10.** Implementation by interference of light. (**a**) Optical micrograph illustration of the experimentally demonstrated OIU [28]. (**b**) Schematic representation of our two-layer ONN experiment [28]. (**c**) Proposed slimmed layer implementation [93]. (**d**) Schematic diagram of a single layer of the proposed architecture [94]. (**a**,**b**) Reprinted with permission from Ref. [28]. Copyright © 2017, Nature Publishing Group. (**c**) Reprinted with permission from Ref. [93], ACM. (**d**) Reprinted with permission from Ref. [94]. Copyright © 2020 IEEE.

### 4.2. Implementation by Resonance of Light

The schemes realized by the resonance of light are noncoherent and can be implemented using microrings for WDM-Matrix–vector multiplication (WDM-MVM) operation. The "broadcast-and-weight" architecture based on the WDM concept was first introduced by Tait in 2014 [11], where optical signals are modulated in parallel and reconfigured by tuning MRs that operate only at specific wavelengths. The WDM concept provides a feasible direction for constructing neuromorphic computing architecture based on microrings since the compactness of microrings can significantly reduce the footprint and improve integration.

"Broadcast-and-weight" tools are constantly used to implement MLP, CNN, and SNN, and several variations of them have been proposed. The "Hitless weight-and-aggregate" architecture that aims to overcome weight corruption caused by thermal factors isolated each weight and tuned them independently [96], as shown in Figure 11a, where they co-designed the microring silicon photonic architectures with FPGA, providing a way to construct large-scale matrix multiplication using MRRs in the wavelength domain and reducing the system decomposition complexity. Combing MRs with memristors to

implement a CNN is a novel approach based on the MR-weight bank [97], in which the convolution layer mainly consists of a weight resistor array, a photonics weight bank, and an SRAM buffer. The MR bank receives weights through memristors and stores them in the SRAM buffer. The Convlight architecture proposed in this work shows a scalable memristor-integrated photonic CNN accelerator, which is the first of its kind. Furthermore, Xu et al., as we discussed before, proposed to use microcombs to enable broad bandwidth convolutional ONNs [8], where the schematic of the photonic CNN is shown in Figure 11b.

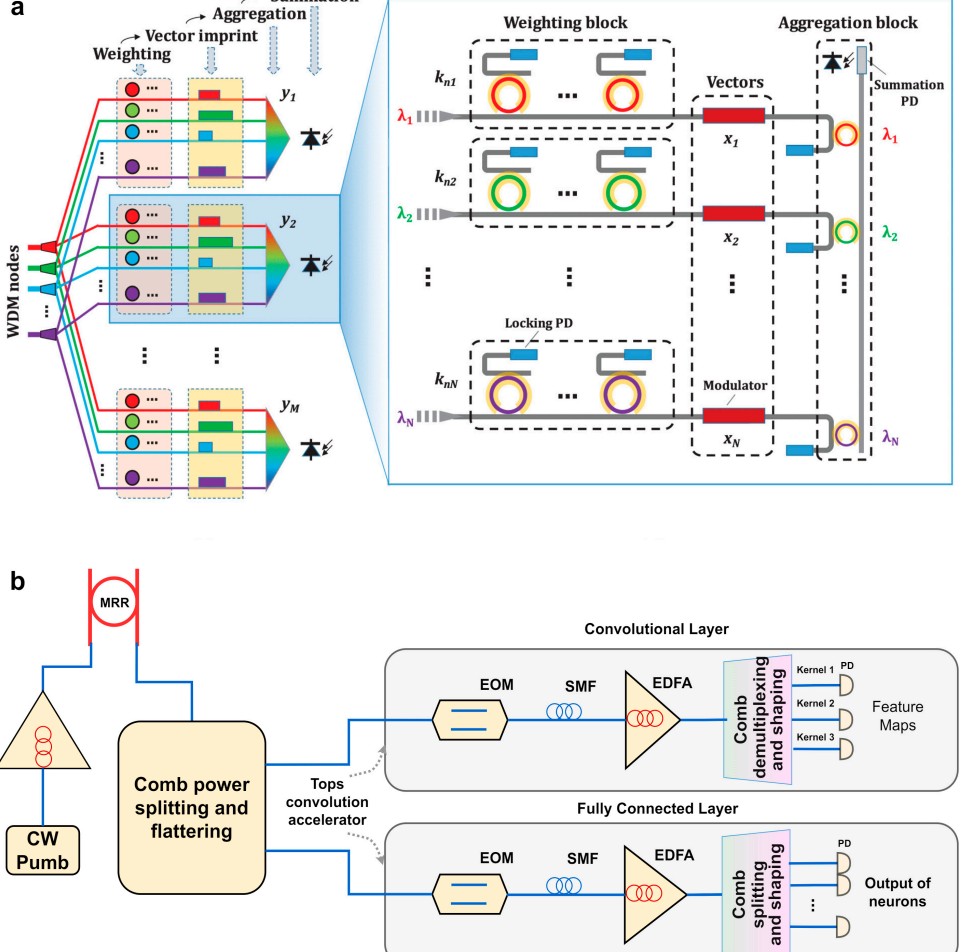

**Figure 11.** Implementation by resonance of light. (**a**) Hitless weight-and-aggregation architecture for M × N vector-matrix multiplier [96]. (**b**) Experimental schematic of the optical CNN [8]. (**a**) Reprinted with permission from Ref. [96]. Copyright © 2020 IEEE. (**b**) Reprinted with permission from Ref. [8]. Copyright © 2021. The author(s), under exclusive license to Springer Nature Limited.

SNN can be implemented with or without the "broadcast-and-weight" protocol. Reference [98] explored the possibility of implementing SNN based on the "broadcast-and-weight" protocol. In addition to the "broadcast-and-weight" architecture, in 2019, Feldmann et al. used microrings integrated with PCM cells to implement an all-optical spiking neurosynaptic network [18], which can control the propagation of light propagation through the ports of PCM cells by simply altering the PCM's state. RC can also be implemented by MRs, for example, in reference [99], where they built a 4 × 4 swirl reservoir on a silicon platform with nodes consisting of nonlinear microporous resonators. An analysis of performance on nonlinear Boolean problems was presented to show the capability of this reservoir, which differs from the conventional swirl-topology reservoir architectures, to set their nodes at near instability.

### 4.3. Algorithm

Previously, most ONN training was achieved by digital electronics, as backpropagation in the silicon photonic domain remains a challenge because nonlinearity in the backward direction must be the gradient of its inverse direction, which remains a challenge. Another problem with online training is that each local optical parameter, such as the intensity of the section, is required to be probed within the integrated photonic circuits.

In order to address the second problem, Hughes et al. proposed the photonic analog of the backpropagation algorithm using adjoint variable methods, as shown in Figure 12c–e [100]. The adjoint variable method is the technology previously utilized in the optimization and inverse design of photonic structures [101], and this training algorithm is effective for the integrated photonic ANNs and other photonic platforms because of its derivation from Maxwell's equations.

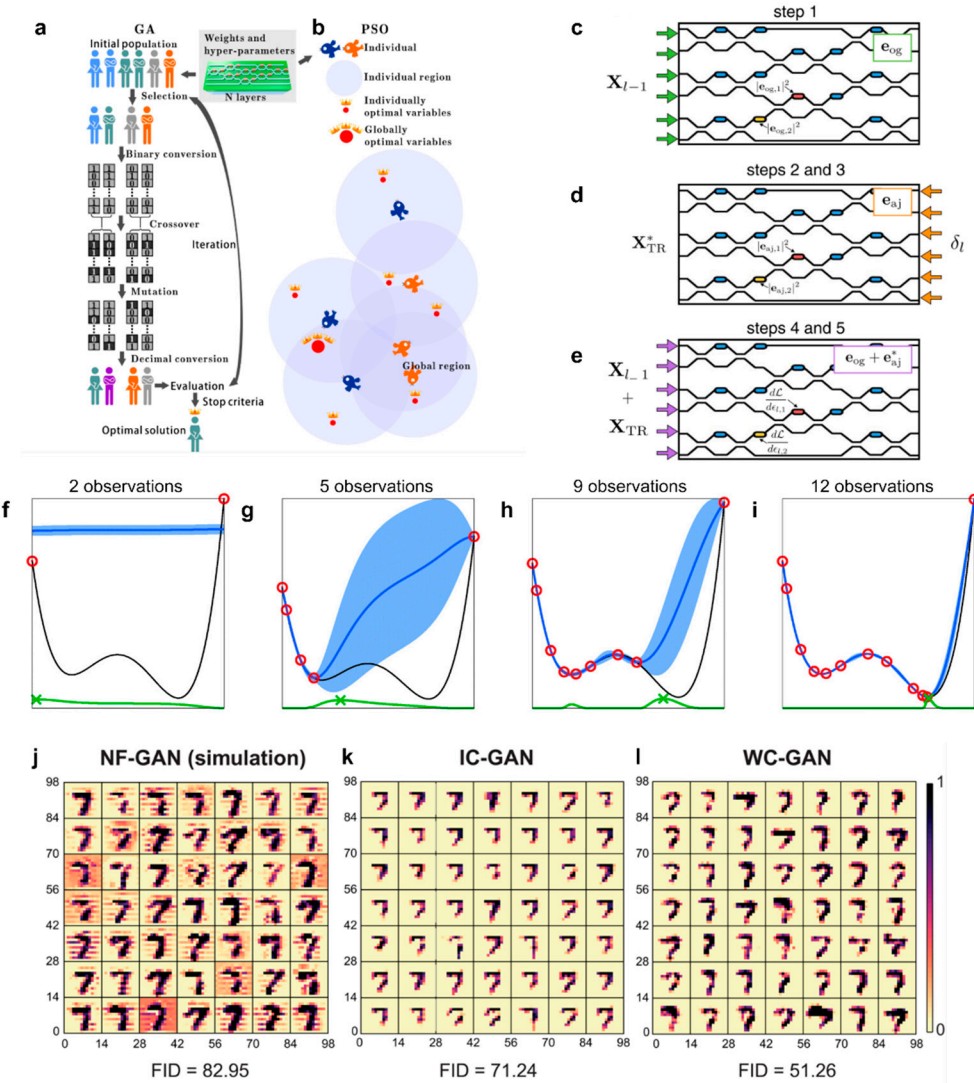

**Figure 12.** Algorithm. (**a**) The flowcharts of the learning algorithms for the ONNs based on GA (genetic algorithm) [102]. (**b**) The flowcharts of the learning algorithms for the ONNs based on PSO (particle swarm optimization) [102]. (**c–e**) Schematic illustration of the proposed method for experimental measurement of gradient information [100]. (**f–i**) Illustration of the Bayesian optimization on a toy 1D problem [103]. (**j–l**) Generating handwritten numbers with GAN. (**j**) to (**l**)

Forty-nine images (size, 14 × 14 pixels) generated by NF-GAN (noise free–trained GAN), IC-GAN (input-compensatory GAN), and WC-GAN (weight-compensatory GAN) [104]. (**a**,**b**) Adapted from Ref. [102]. CC BY 4.0. (**c**–**e**) Adapted from Ref. [100]. CC BY 4.0. (**f**–**i**) Reprinted with permission from Ref. [103]. Copyright © 2021, Springer Science Business Media, LLC, part of Springer Nature. (**j**–**l**) Adapted from Ref. [104]. CC BY 4.0.

Statistical optimization tools can be alternatives to backpropagation; for example, genetic algorithm and particle swarm optimization were used to train the hyperparameters and optimize the weights in reference [102], as shown in Figure 12a,b. These two algorithms are both gradient-free and show effectiveness when used for classification tasks with different datasets in a trained ONN. Bayesian optimization has also been used in reservoir computing in several simulations [103] because Bayesian optimization also provides a better grasp of the significance of various hyper-parameters. Figure 12f–h,i show the Bayesian optimization on a toy 1D problem.

In RC networks, nonlinearity inversion can be realized because, in the training methods [105], the reservoir's states are estimated through a single photodetector at its output, which includes an approximate inversion of the nonlinearity of photodetectors. This method solves the lost ability to observe the condition of reservoirs by photodetectors (because in all-optical RC networks, fewer photodetectors are required, which receives the weighted total of all the optical signals), which is necessary for many linear training algorithms. Moreover, as we discussed, STDP is a crucial and popular synaptic weight plasticity model used to simulate the synaptic plasticity between neurons, and thus unsupervised learning using the STDP mechanism for training spiking neural networks (SNNs) is the most common in recent work.

Moreover, how to reduce and resolve inherent optoelectronic stochasticity and non-ideality are both key issues for photonic integrated circuits. On the one hand, the accumulation of errors in photonic systems due to their analog nature and the abundance of optoelectronic noise has the potential to drastically affect their performance; on the other hand, manufacturing variations and thermal crosstalk limit their practicality and performance. Hence, Wu et al. proposed a photonic Generative adversarial network (GAN) and the corresponding noise-aware training approaches [104]. After training with noise-aware training methodologies, namely, the input-compensatory approach (IC-GAN) and the kernel weight-compensatory approach (WC-GAN), the photonic generative network may not only withstand but also profit from a certain degree of hardware noise, as shown in Figure 12j–l. Previous research also showed several offline noise-aware training schemes, including injecting noises into layer inputs, synaptic weights, and preactivation [106]. Furthermore, the first self-calibrated photonic chip was demonstrated by Xu and his colleagues [107]. As we discussed, self-calibration is imperative in both electronics and photonics because it could guarantee a stable performance of devices. The self-calibration technique incorporating an optical reference path into the PIC used the Kramers–Kronig relationship to recover the phase response from amplitude measurements and achieved a fast-converging self-calibration algorithm.

In addition to the methods above, forward propagation in ONNs is much easier to implement and can be computed in a constant time with very low power consumption. Although it is simple in form and convenient in use, only in some simple ONNs or very deep RNNs, it can be executed at a faster rate.

## 5. Outlook and Discussion

In this paper, we reviewed the recent advances achieved in neuromorphic computing based on silicon photonics in detail. An overview of micro-architecture functionalities, devices, architectures, and algorithms in neuromorphic computing based on silicon photonics is displayed. Silicon photonics, which has been explored for a long time, shows great promise for implementing neuromorphic computing since it offers sufficient integration and maturity for the current photonic computing. Hence, neuromorphic silicon photonics is an emerging field that combines the speed and parallelism of photonics with

the adaptiveness of deep learning, which can be theoretically orders of magnitude ahead of traditional electronics. The utilization of new concepts such as WDM, novel devices such as PCM, soliton microcombs and metasurfaces, viable fabrication techniques, and advanced algorithms can allow for unprecedented developments in the next generation of optical neural networks. Here, we summarized the current challenges and pointed out the possible opportunities for the materialization of future optical neural networks.

**Electronic–photonic co-design:** In the absence of electronic controllers, the electronic–photonic co-design neural network is a more practical route for current ANNs until the efficiency of electronic control can be found as a competitive candidate in the optical domain. Although monolithic fabrication provides excellent opportunities to integrate electronics and photonics on the same substrate, the high latencies and power consumption caused by the electronic components pose challenges for electronic controllers. In ONNs, the controller should manage photonic devices and maintain stable operations of neurons in real-time, at high speed and efficiency.

**Lasers:** There is an essential requirement of light generation for an optical computing system. Light sources are required for supplying the modulated input signals or carrying out the nonlinear activation in optical computing, from straightforward operations such as MVM to complex computations such as AI algorithms. However, at face value, silicon appeared to be a poor choice for applications such as light sources, modulators, photodetectors, etc., because of its low electro-optic (EO) coefficients and indirect bandgap. Thanks to an abundance of research and development from both academia and the industry, on-chip silicon lasers were developed by using hybrid integration, heterogeneous integration based on wafer bonding, and monolithic integration based on direct epitaxial growth. Alternatively, due to the inherent inhomogeneously broadened gain profiles of QD lasers [108], they can achieve the coveted microcomb emission with high-power flat-top spectra, making them ideally suited for WDM architectures [109].

**On-chip integration:** Note that on-chip ONNs are the mainstream in current research because mature CMOS technology has enormous advantages for large-scale and highly integrated ONNs. However, the cost of on-chip optical networks is extremely expensive in terms of money overhead, labor expenses, technology requirements, and so on. For instance, coherent architectures which are promising to be fabricated in an integrated way are hampered by the issues with MZIs, due to the large area requirement and phase-noise corruption. In addition, on-chip integration suffers from lifetime instability due to thermal crosstalk and manufacturing process variations for many other photonic devices such as MRs and PCMs. Moreover, silicon on-chip lasers are more susceptible to environmental factors and remain an obstacle in academia.

**Training:** As we discussed before, the training process is usually completed on digital computers in many works. On the one hand, it is crucial to devise an effective training method for current ONNs which can work in the optical domain in real time. On the other hand, exploring photonic architectures that can efficiently support neural network training is promising but challenging, as backpropagation imposes additional requirements on the current photonic neural networks. The training of networks based on scattering and diffraction of light can be a reference for silicon-based neuromorphic photonics. Recent work using metasurfaces to implement diffractive neural networks has taken an important step and shown possibilities for future training methods [32]. Moreover, the photonic accelerator based on memristors hybrid hardware supports backpropagation [110], which can also be a great example.

**Scalability:** Scalability is the most evident problem between ONNs and electronic ANNs. The advances achieved in ONNS are indelible, but the issues are that many works focus on small-scale ONNs compared to electrical ANNs, which can have millions of weight parameters. The most practical approach to addressing this problem is to optimize and improve optical components. On the other hand, more structures suitable for optical neural networks are required to be proposed to reduce the complexity of the network; hence, they could pave the way to scale the photonic networks.

On the horizon in the future, there are numerous difficulties to be addressed. Nonetheless, photonic neural networks have found applications in many fields beyond the reach of traditional computer technology, such as intelligent signal processing, high-performance computing, nonlinear programming, and controlling, and these promising prospects motivate people to explore the future of optical neural networks further.

**Author Contributions:** Conceptualization, R.X.; writing—original draft preparation, B.X.; writing—review and editing, B.X., Y.H., Y.F., Z.W., S.Y. and R.X. All authors have read and agreed to the published version of the manuscript.

**Funding:** This research was funded by the Key R&D program of China (2021ZD0109904) and Start-up fund from the Hong Kong University of Science and Technology (Guangzhou).

**Institutional Review Board Statement:** Not applicable.

**Informed Consent Statement:** Not applicable.

**Data Availability Statement:** Not applicable.

**Conflicts of Interest:** The authors declare no conflict of interest.

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
