# Peer review of "Recent Progress of Neuromorphic Computing Based on Silicon Photonics: Electronic–Photonic Co-Design, Device, and Architecture"

_photonics, doi:10.3390/photonics9100698_

Round 1
Reviewer 1 Report
The authors review recent progress of Neuromorphic Computing based on Silicon Photonics from three aspects: Electronic-Photonic Co-Design, Device and Architecture. The manuscript has exhaustively covered most recent advances in neuromorphic computing based on silicon photonics. My major comments are as follow:
1. Line 177, VSCEL should be VCSEL?
2. Line 331, “A single” should be “a single”?
3. Line 390, 3.3.2 should be 3.2.2?
4. In section 3.2.2 (Line 390), the authors mainly review devices fabricated by depositing different PCMs on silicon photonic structures like ring resonator, waveguide, etc. I wonder if there is another configuration of PCM devices that can also perform neuromorphic computing such as polarization-selective nanowires based barely on silicon substrate and GST (J. S. Lee, et al. Sci. Adv. 8, eabn9459, (2022)). A more comprehensive discussion may be helpful.
5. Line 511, the authors claim that phase delays are created by altering the size of silicon slots. As far as I know, the phase delays can also be created by PCM arrays fabricated on top of the Si3N4 waveguide, which can be utilized to demonstrate a prototypical optical convolution neural network (C. Wu, et al. Nat. Commun. 12, 96 (2021)). Therefore, can authors make a more comprehensive discussion here with different methods of creating phase delays?
Reviewer 2 Report
This work proposes an extensive review on recent advances achieved in neuromorphic computing based on silicon photonics, including the electronics-photonics co-design, devices, architectures, and algorithms in neuromorphic computing. The review is of interest and well written, I recommend for it to be published in the present form. Here are some minor comments the authors should consider addressing. E.g., a reference need to be added in line 129, page 7; References [101-108] are only shown in the figure caption while not in the in the main text. I suggest authors carefully check all the references shown in the figure captions.
